

# Urinary aluminium and its association with autism spectrum disorder in urban preschool children in Malaysia

Mohd Shahrol Abd Wahil[1,2], Mohd Hasni Ja'afar[1] and Zaleha Md Isa[1]

[1] Department of Community Health, Faculty of Medicine, Universiti Kebangsaan Malaysia, Cheras, Kuala Lumpur, Malaysia
[2] Sector of Occupational and Environmental Health, Disease Control Division, Ministry of Health Malaysia, Putrajaya, Malaysia

## ABSTRACT

**Background**. The presence of aluminium (Al) in the human body may impact brain neurodevelopment and function, and it is thought to contribute to autism spectrum disease (ASD). The main objective of this study was to assess the association between urinary Al and the development of ASD among Malaysian preschool children in the urban city of Kuala Lumpur.

**Method**. This was an unmatched case–control study in which children with ASD were recruited from an autism early intervention center and typically developed (TD) children were recruited from government-run nurseries and preschools. Urine samples were collected at home, assembled temporarily at study locations, and transported to the laboratory within 24 h. The Al concentration in the children's urine samples was determined using inductively coupled plasma mass spectrometry (ICP-MS).

**Result**. A total of 155 preschool children; 81 ASD children and 74 TD children, aged 3 to 6 years, were enlisted in the study. This study demonstrated that ASD children had significantly higher urinary Al levels than TD children (median (interquartile range (IQR): 2.89 (6.77) µg/dL versus 0.96 (2.95) µg/dL) ($p < 0.001$). Higher parental education level, non-Malay ethnicity, male gender, and higher urinary Al level were the significant ASD risk factors (adjusted odds ratio (aOR) $>1$, $p < 0.05$).

**Conclusion**. A higher urine Al level was discovered to be a significant risk factor for ASD among preschool children in the urban area of Kuala Lumpur, Malaysia.

## INTRODUCTION

Autism spectrum disorder (ASD) is an early-onset neurodevelopmental condition characterized by altered social interaction and communication, as well as repetitive and rigid behavior, together with restricted interests (*APA, 2013*). It has the potential to substantially affect major aspects of life and lower quality of life beginning in childhood and continuing into later adulthood (*Bölte, Girdler & Marschik, 2019*). Individuals with ASD may experience the aforementioned characteristics to variable degrees and demonstrate

Corresponding author
Mohd Hasni Ja'afar,
drmhasni@ukm.edu.my

a broad range of abilities, which can range from being extremely gifted to being severely challenged (*Freedman, 2010*).

ASD prevalence has significantly grown globally from its first description in 1943, when there were 4.5 cases per 10,000 children (*Kanner, 1943*). For instance, the prevalence of ASD in the United States grew from 3.4 cases per 1,000 children in the early 2000s (*Fombonne, 2003*) to 20.0 cases per 1,000 children in 2011 (*Blumberg et al., 2013*). In the United Kingdom, the prevalence of ASD in 2009 was 9.4 cases per 1,000 children (*Baron-Cohen et al., 2009*). While the prevalence of ASD in Malaysia in 2011 was 15.9 cases per 1,000 children (*Kaur et al., 2015*).

Aluminium (Al) is the most ubiquitous metal and the third most abundant element in the Earth's crust (*Exley, 2009*). It has no known biological role in human and is not essential at any quantity (*Rahbar et al., 2015*). However, its distinguishing qualities, such as lightweight, strength, and ease of sterilization, make it suitable for use in pharmaceuticals, water, food, and other household items (*Corkins, 2019*). In view of its widespread usage, humans are at high risk of exposure either through oral ingestion, skin absorption (wounds), or occupational inhalation (*ATSDR, 2008*). Once Al enters the circulation and tissue fluids, it swiftly equilibrates with chemically similar elements in the body and distributes broadly (*Priest, 1993*). Only the skeleton contains a high concentration of Al, accounting for 54.0% of the body's total Al content (*Corkins, 2019*). Aside from skeleton, one of the most important target tissues for Al accretion is the brain (*Bondy, 2016*), which may penetrate the blood–brain barrier (BBB) *via* particular carrier protein complexes (*Yokel et al., 2001*; *Yokel et al., 2002*). Al is predominantly eliminated from the body through urine and feces in humans (*ATSDR, 2008*).

Al intoxication in humans has been shown to have impacts on the musculoskeletal system, the hematological system, and the neurological system (*Crisponi et al., 2013*). Studies have shown that the neurotoxic effect of Al is caused by oxidative stress, which leads to cell death (apoptosis) through; (1) interactions with transitional metals like copper and iron (*Becaria, Bondy & Campbell, 2003*; *Di & Bi, 2004*), (2) inhibition of cytochrome oxidase (in the case of Al phosphate (AlP) toxicity) (*Chugh et al., 1993*), (3) increased lipid peroxidation (*Ferretti et al., 2003*; *Verstraeten, Aimo & Oteiza, 2008*), (4) enhancer of 6-hydroxydopamine (*Sánchez-Iglesias et al., 2009*), and (5) generation of reactive oxygen species (ROS) in electron transport chain (*Campbell, 2002*; *Kumar & Gill, 2014*). Additionally, Al complexing changes several features of BBB permeability in such a way that an Al-amyloid beta complex would be more readily penetrate to brain cells than amyloid beta alone (*Al-Ayadhi et al., 2012*; *Banks et al., 2006*).

Children are considered a vulnerable population because they may be more susceptible to health consequences than adults, and the link may vary depending on developmental stage (*Bearer, 1995*). Compared to adults, children might also have a different capacity for repairing damage from neurotoxicant insults and have a longer remaining lifetime in which neurotoxicant damage can be expressed (*Rice & Barone Jr, 2000*). In addition, both the prenatal and postnatal periods are crucial for anatomical and functional brain development, in which exposure to neurotoxicant in these phases might impair the brain more significantly (*Gale et al., 2004*). Furthermore, the health consequence that we are most

concerned about in children is that the harm may not appear until a later developmental stage (*WHO, 2006*).

According to past research, Al is linked to neuropathological abnormalities seen in neurodegenerative illnesses in adults such as Alzheimer's disease, Parkinson's disease, amyotrophic lateral sclerosis, and multiple sclerosis (*Bondy & Campbell, 2017*). Nonetheless, toxicological research on the effects of Al on neurodevelopmental disorders in young children, particularly ASD, is scarce. Besides that, the statistical profile of Al in humans, particularly children, in relation to ASD in Malaysia is unavailable. In view of the foregoing, the primary goal of this study was to examine the relationship between urinary Al and ASD among preschoolers in Kuala Lumpur, Malaysia.

## MATERIAL & METHODS

This was an unmatched case-control study involving 155 preschool children, consisting of 81 ASD children and 74 TD children (age 3 to 6 years) in the urban city of Kuala Lumpur in the year of 2019. The ASD kids were selected at random from the Genius Kurnia center in Kuala Lumpur, which is a government center for early autism intervention. The TD children were also randomly selected from Tabika Kemas public preschools and Taska Kemas public nurseries in Kuala Lumpur. The study sites for case and control group were also randomly selected to maintain the random effect in the study. The pediatrician's diagnosis was according to Diagnostic and Statistical Manual of Mental Disorders fifth edition (DSM-5) criteria (*APA, 2013*) and the International Classification of Diseases tenth revision (ICD-10) (*WHO, 2004*). As per the Modified Checklist for Autism in Toddlers (M-CHAT) assessment, the TD children were declared healthy by the pediatrician. For both groups, the following exclusion standards are applied; (1) other neurobehavioral disorders, (2) congenital abnormality or syndrome, (3) acute infectious diseases, (4) recent surgical procedures, (5) endocrine problems, and (6) on oral/infusion drugs (chemotherapy) or chelation treatment for eliminating heavy metals. The selection of the respondents is illustrated in Fig. 1.

The evaluation included the participation of both the parents and the children. An online questionnaire that the parents were supposed to fill out on their own time was sent to them. The children were obliged to provide clean-catch urine in the morning. Urine samples were obtained once by the parents at their residence and then handed over to the investigator on the same day that their children started attending classes (at the Genius Kurnia center, Tabika Kemas preschools, and Taska Kemas nurseries). The urine samples were then transported within 24 h to an environmental laboratory, located 40.0 km from the study locations at the National University of Malaysia (UKM), Bangi, Selangor state. These samples were stored at $-20.0\ ^{\circ}$C before laboratory analysis. The urine samples were then treated with a 0.2% nitric acid ($HNO_3$) solution to facilitate the breakdown of organic particles in the urine sample (*Abd Wahil, Ja'afar & Isa, 2021*), followed by Al assay using inductively coupled plasma mass spectrometry (ICP-MS) on a PerkinElmer SCIEX™ ELAN® 9000. The system was calibrated, and internal standardization was conducted to ensure good performance of the system in term of sensitivity and accuracy. The Al

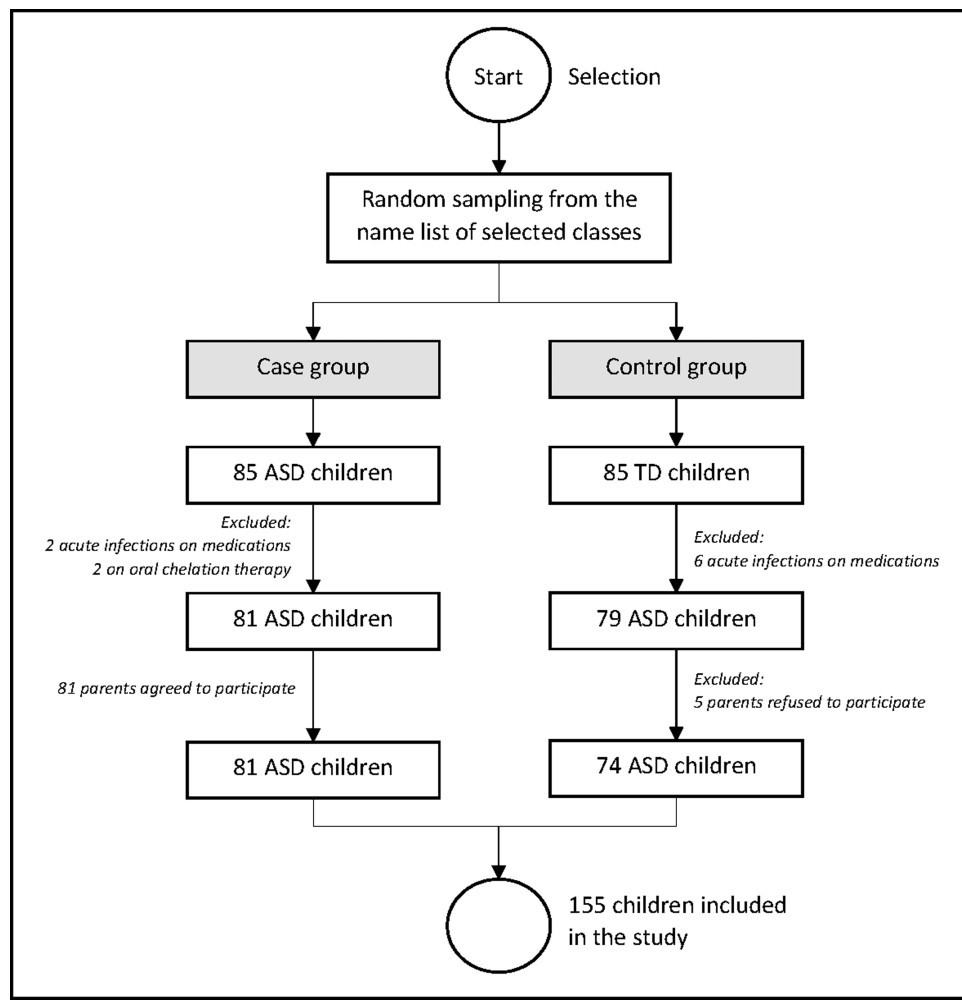

**Figure 1  Selection of the respondents into the study.**

detection limit was from 1.0 to 10.0 parts per trillion (ppt). The concentration readings of Al were obtained from computer software linked to the ICP-MS machine. Finally, the ICP-MS results and questionnaire data set were analyzed using the software of International Business Machines (IBM), Statistical Package for Social Sciences (SPSS), version 22.

The normality of the data was examined graphically using Q-Q plot and histogram and statistically by Kolmogorov–Smirnov statistics, kurtosis, skewness, mean and median. Due to the non-normal distribution of the data, the non-parametric tests such as Mann–Whitney $U$ test and Kruskal-Wallis test were used to measure the comparison of Al levels between groups. Multiple logistic regression was performed to assess the association between urinary Al and the development of ASD among preschool children.

The Ministry of Health (MOH) Malaysia Medical Research and Ethics Committee (NMRR-18-3765-45117) and the National University of Malaysia (UKM) Research and Ethics Committee (FF-2018-286) approved the study in 2019. All processes were carried out in accordance with the principles of the Helsinki Declaration of 1964 and

its amendments. Prior to the study, parents provided informed written agreement and participated voluntarily.

## RESULTS

A total of 155 individuals were recruited for this study, with 81 ASD children in the case group and 74 TD children in the control group. The demographic characteristics are shown in Table 1. From this table, there was a significant difference in composition between the two groups in terms of parental education level, family monthly income, ethnicity, and children's gender. Most of the ASD children's parents had tertiary education (85.2%), however, most of the TD children's parents had secondary education (60.8%) ($p < 0.001$). The ASD families had an average monthly income of RM 6,699.94 ($\pm$ RM 3,856.52) while that of the TD families was RM 3,960.03 ($\pm$RM 2,025.19) ($p < 0.001$). About 51.9% of the ASD families had a monthly income of >RM 5,000, whereby, only 21.6% of the TD families had a monthly income of >RM 5,000 ($p < 0.001$). Among the ASD and TD children, 77.8% and 94.6% were Malay, respectively ($p = 0.003$). Male gender was predominant among the ASD children (84.0%) with male: female ratio of 5:1, while TD group had an almost equal proportion of boys and girls ($p < 0.001$). The average age of the ASD and TD children was $5.63 \pm 0.60$ years and $5.45 \pm 0.83$ years, respectively ($p > 0.05$).

In Table 2, the concentration of urinary Al was compared between ASD children and TD children. The ASD children were found to have a significantly higher median (interquartile range (IQR)) urinary Al level than the TD children (2.89 (6.77) μg/dL *versus* 0.96 (2.95) μg/dL) ($p < 0.001$). Furthermore, the ASD children also had higher level of minimum, maximum, mean ($\pm$SD), and geometric mean urinary Al than the TD children. When ASD children were classified according to the severity of the disease, urinary Al levels were surprisingly increased in a trend with the severity of the ASD ($p < 0.001$) (Fig. 2).

The receiver operating characteristic (ROC) curve analysis was performed to determine the urinary Al level cut-off. The urinary Al area under the curve had a significant value of 0.75 and close to the value of 1.0 (Fig. 3). The value of 1.0 shows that the model is perfect. According to the Youden index, the cut-off level of urinary Al was 1.60 μg/dL, higher than the standard reference of 0.58–0.77 μg/dL from the report of the Agency for Toxic Substances and Disease Registry (*ATSDR, 2004*). This new cut-off level of urinary Al was used to categorize urinary Al into 2 groups (>1.6 μg/dL and $\leq$ 1.6 μg/dL) in the regression analysis.

Table 3 illustrates the findings of the multiple logistic regression analysis for the ASD-associated factors. Parental education level, ethnicity, children's gender, and urinary Al level were significant ASD risk factors. When compared to parents with secondary education, parents with tertiary education had 17.9 times the odds of having ASD children (adjusted odds ratio (aOR) = 17.89; 95.0% confidence interval (CI): 5.97–53.63; $p < 0.001$). Compared to Malay children, non-Malay children had 6.1 times greater odds of having ASD (aOR = 6.11 ; 95.0% CI [1.17–31.96]; $p = 0.032$). Compared to girls, boys had 9.8 times greater odds of having ASD (aOR = 9.82; 95.0% CI [3.22–30.00]; $p < 0.001$). Compared to children with urinary Al levels of $\leq$1.6 μg/dL, children with urinary Al levels of >1.6

**Table 1  Respondents' characteristics for both case and control groups.**

| Variables | ASD ($n = 81$) $n$ (%) | TD ($n = 74$) $n$ (%) | $x^2$-value | $p$-value |
|---|---|---|---|---|
| Parental Background | | | | |
| Education Level | | | | |
|     Secondary Education | 12 (14.8) | 45 (60.8) | 35.187 | |
|     Tertiary Education | 69 (85.2) | 29 (39.2) | | <0.001[*] |
| Mean (± SD) Monthly Income in Ringgit Malaysia (RM) | 6699.94 ± 3856.52 | 3960.03 ± 2025.19 | −5.461[#] | <0.001[*] |
| Median (IQR) Monthly Income in (RM) | 5500.00 (6500.00) | 3500.00 (2575.00) | −4.867[ε] | <0.001[*] |
| Income Classification | | | | |
|     ≤RM5000.00 | 39 (48.1) | 58 (78.4) | 17.922 | |
|     >RM5000.00 | 42 (51.9) | 16 (21.6) | | <0.001[*] |
| Children Background | | | | |
| Ethnicity | | | | |
|     Malay | 63 (77.8) | 70 (94.6) | 8.980 | |
|     Non-Malay | 18 (22.2) | 4 (5.4) | | 0.003[*] |
| Gender | | | | |
|     Male | 68 (84.0) | 39 (52.7) | 17.663 | |
|     Female | 13 (16.0) | 35 (47.3) | | <0.001[*] |
| Mean (± SD) Age (Years) | 5.63 ± 0.60 | 5.45 ± 0.83 | −1.188[#] | 0.114 |
| Median (IQR) Age (Years) | 6.00 (1.00) | 6.00 (1.00) | −1.163[ε] | 0.245 |
| Age Group | | | | |
|     ≤4 years old | 5 (6.2) | 10 (13.5) | 2.384 | |
|     >4 years old | 76 (93.8) | 64 (86.5) | | 0.123 |

Notes.
[*]$p < 0.05$ indicates significant statistical result.
[#]$t$-value from Student's $T$-Test to determine comparison of mean (SD) of the continuous variables between ASD and TD group.
[ε] $z$-value from Mann–Whitney $U$ Test to determine comparison of median (IQR) of the continuous variables between ASD and TD group.

**Table 2  Comparison of the urinary Al concentration between case and control groups.**

| Urinary Al level | ASD | TD | $p$-value[#] |
|---|---|---|---|
| No. of Respondents (%) | 81 (52.26%) | 74 (47.74%) | |
| Minimum (μg/dL) | 0.04 | 0.001 | |
| Maximum (μg/dL) | 47.24 | 24.10 | |
| Mean (±SD) (μg/dL) | 5.06 (± 6.30) | 2.58 (± 4.89) | |
| Geometric Mean (μg/dL) | 2.93 | 0.27 | |
| Median (IQR) (μg/dL) | 2.89 (6.77) | 0.96 (2.95) | <0.001[*#] |

Notes.
[*]$p < 0.05$ indicates significant statistical result.
[#]Comparison of median (IQR) urinary Al concentration between ASD and TD groups using Mann Whitney $U$ Test.

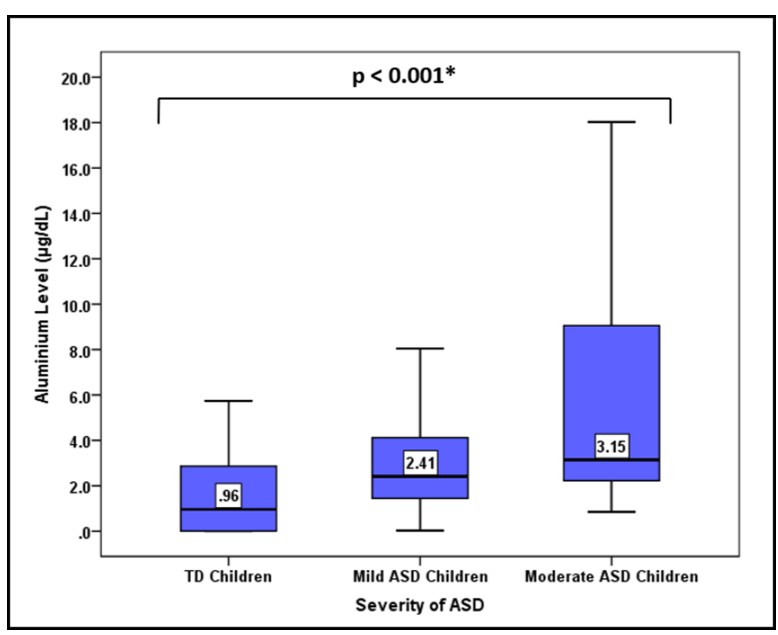

**Figure 2 Comparison of the median urinary Al concentration according to the severity of ASD.** *Comparison of median urinary Al concentration between degree of disorder severity using Kruskal Wallis test.

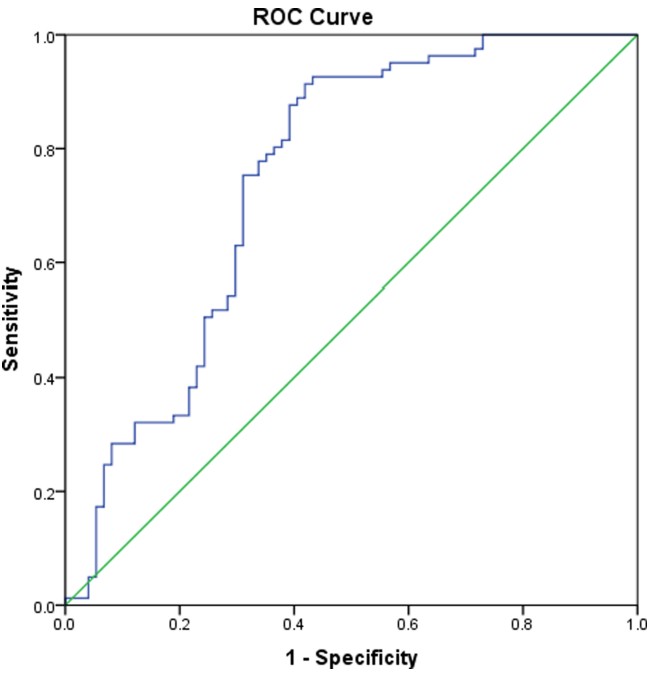

**Figure 3 Receiver operating characteristic (ROC) curve analysis of urinary Al and ASD.**

**Table 3  Multiple logistic regression analysis of associated factors for ASD.**

| Variables | B | Wald | aOR | 95.0% CI | p-value |
|---|---|---|---|---|---|
| Parental Background | | | | | |
| Education Background | | | | | |
|    Tertiary Education | 2.88 | 26.52 | 17.89 | 5.97, 53.63 | <0.001[*] |
|    Secondary Education | | | 1.00 | | |
| Children Background | | | | | |
| Ethnicity | | | | | |
|    Non-Malay | 1.81 | 4.60 | 6.11 | 1.17, 31.96 | 0.032[*] |
|    Malay | | | 1.00 | | |
| Gender | | | | | |
|    Male | 2.29 | 16.08 | 9.82 | 3.22, 30.00 | <0.001[*] |
|    Female | | | 1.00 | | |
| Heavy Metal | | | | | |
| Urinary Al | | | | | |
|    >1.6 μg/dL | 2.55 | 22.88 | 12.77 | 4.50, 36.27 | <0.001[*] |
|    ≤1.6 μg/dL | | | 1.00 | | |

Notes.
  [*]$p < 0.05$ indicates significant statistical result from multiple logistic regression using stepwise method.
  aOR, adjusted odds ratio; CI, confidence interval.

μg/dL had 12.8 times greater odds of having ASD (aOR = 12.77 ; 95.0% CI [4.50–36.27]; $p < 0.001$).

# DISCUSSION

The long-lasting and permanent human health effects of environmental neurotoxins have become a great concern nowadays. It is due to the hazard characteristic of neurotoxicants, which impair physiological function of organs at low concentrations and cause permanent damage to the organ's morphological features. The presence of neurotoxicants in environmental media (such as air, water, soil, food, and household products), particularly in highly polluted areas such as urban cities and industrial zones, would increase the population's risk of exposure. Young children, in particular, have more potential to receive the effects of neurotoxicants than adults due to their immature organs' ability to detoxify the hazardous chemicals in the body and develop important organs such as the brain. The exposure to neurotoxins could increase due to their mouthing behavior and curiosity (they tend to explore and put anything into their mouth), and food selectivity (they are picky eaters, which leads to less intake of essential nutrition).

Children's being exposed to neurotoxicants, particularly Al at the critical developmental stages (less than 6 years), may be related to the etiology of ASD. Different magnitudes of exposure might affect the spectrum of the disease. We were able to demonstrate the significant pattern of the autism spectrum in relation to urinary Al concentration. In this study, there was a concrete finding between ASD severity and the accumulation level of Al in the urine. Here, children with moderate-functioning ASD had the highest concentration level of urinary Al, followed by that in children with high-functioning ASD and TD

children. This indicates higher Al exposure among children with more severe ASD. To the current level of our knowledge, no study has ever demonstrated similar findings pertaining to urinary Al and ASD. However, a study done in India revealed a significant elevation of other neurotoxicants, lead (Pb) and mercury (Hg) in the hair and nail samples as ASD becomes more severe (*Priya & Geetha, 2011*).

Fundamentally, we were able to demonstrate that ASD children had greater urine Al concentrations than TD children. This finding was supported by previous studies. For instance, a study demonstrated significantly increased urinary Al levels in 55 Egyptian ASD children ($34.56 \pm 20.88$ µg/g creatinine) compared to 75 TD children matched by age, gender, and socioeconomic status ($17.56 \pm 5.75$ µg/g creatinine) (*Metwally et al., 2015*). Besides that, a study was done on 25 ASD children and 25 TD children matched by age and gender in Egypt and reported that the ASD children had significantly higher mean urinary Al ($111.26 \pm 142.94$ µg/g creatinine) than the TD children ($22.98 \pm 21.29$ µg/g creatinine) (*Blaurock-Busch, Amin & Rabah, 2011*).

To date, no report has published a cut-off point for urinary Al level. It is important to determine the safe level of Al for the reference in the human biomonitoring program. In 2004, the Florida Department of Health reported a wide range of urinary Al of 0.50 µg/dL to 5.50 µg/dL among children in Suwanne County, Florida (*ATSDR, 2004*). In this study, we were able to determine the cut-off point of urinary Al of 1.60 µg/dL among pre-school children in the urban city of Kuala Lumpur. It should be noted that the proposed cut-off point was based on small sample size, and it has not been validated on an independent dataset. However, this value may be useful for comparison in future studies conducted among children living in other major cities in Malaysia, such as Johor Bahru (in Johor state), Klang (in Selangor state), Georgetown (in Penang state), Kuching (in Sarawak state), and Kota Kinabalu (in Sabah state).

The regression analysis in this study demonstrated urinary Al concentration as an important risk factor for developing ASD among young children. Children with a urinary Al concentration of >1.60 µg/dL had higher odds of developing ASD than children with a urinary Al concentration of ≤1.60 µg/dL. In other words, higher urinary Al levels reflect higher exposure to Al. A study done in Japan demonstrated higher urinary Al concentration among Al-handling workers (median: 20.10 µg/g creatinine) than non-Al-handling workers (median: 8.80 µg/g creatinine) (*Ogawa & Kayama, 2015*). Another study done in Sweden demonstrated the highest urinary Al concentrations among the exposed workers (median 8.10 µmol/L) and lower concentrations among retired workers (median 0.80 µmol/L) and occupationally non-exposed workers (median 0.09 µmol/L) (*Ljunggren, Lidums & Sjögren, 1991*). The higher body burden of Al eventually impaired the function of the brain and could be linked to neurobehavioral diseases, particularly ASD (*Mold et al., 2018*; *Morris, Puri & Frye, 2017*).

The findings from this study also indicate that the assessment of Al in relation to ASD can be done through urine sample analysis and the finding is reliable. Besides that, urine collection is preferable among young children over blood sampling due to its non-invasive procedure. Assessment of Al in the urine may indicate acute exposure (*Xu, Pai & Melethil, 1991*). However, it may also recur over a long period as a result of frequent exposure through

oral ingestion and inhalation. Nonetheless, elimination of Al may require years and could be due to Al retention in body storage from which it is removed progressively (*Priest et al., 1995*). Most likely, this storage occurs in the bone, which contains approximately 60.0% of the human Al body burden (*Krewski et al., 2007*). It is also predicted that slow Al removal accompanied by continued exposure would produce an increasing body burden with age (*Yokel & McNamara, 2001*). In other words, a long-term exposure to Al gives rise to an accumulation of Al in the body and skeleton of healthy persons, and that the elimination of retained Al is very slow, in the order of several years (*Elinder et al., 1991*; *Ljunggren, Lidums & Sjögren, 1991*). The results of this study and the evidence listed above can be used as reference and supporting evidence upon the proposal and development of human biomonitoring program in Malaysia.

The main strategies for reducing exposure to Al should include the environmental health approach. The government should strengthen preventive measures for controlling and monitoring Al levels in food and water. For instance, currently, heavy metal contaminants in food products are monitored by the Food Safety and Quality Division, MOH Malaysia, while the water quality is controlled and monitored by the water supply concessionaire and the Drinking Water Quality Control Unit, MOH Malaysia. Instant action should be taken when Al concentrations in food and drinking water exceed safe levels, for example, by withdrawing the product from the market and halting the supply of public drinking water until remedial action is taken.

While environmental monitoring of pollutants in specific environmental media (air, water, soil, food, and domestic products) is important for the control and prevention of exposure, it cannot reflect the real exposure received by humans. Many developed countries have implemented human biomonitoring to monitor human exposure to environmental pollutants (*WHO, 2015*). Human biomonitoring allows us to measure exposure to pollutants by measuring either the substances themselves, their metabolites, or markers of subsequence health effects in body fluid or tissues. At present, there is no established human biomonitoring program to monitor the exposure of environmental toxicants among the Malaysian population. Therefore, the MOH Malaysia should introduce the first ever human biomonitoring program in Malaysia in order to monitor the body burden of environmental toxicants including Al among the population. The program could be initiated in high-risk areas such as urban cities and industrialized zones. In terms of sensitive receptors, vulnerable groups such as children, pregnant women, and the elderly should be considered and included in the monitoring program. The monitoring of environmental toxicants can be done through urine samples compared to blood/plasma samples among children in view of the non-invasive procedure, easy to collect, non-expensive, and requiring fewer human resources. The determination of Al in urine is preferred for conducting biological monitoring due to its better sensitivity and robustness when compared to the detection of Al in plasma (*Ogawa & Kayama, 2015*). To run this program, MOH Malaysia should be ready in term of financial support, laboratory capacity and human resources.

There are limitations to this study, and the results should be interpreted with care. Only Al was tested in urine, which may not fully explain the complicated pathogenic

pathways in the brain caused by this neurotoxicant. Moreover, the authors did not include other neurotoxicants like organophosphate pesticides, heavy metals (such as mercury (Hg), lead (Pb), cadmium (Cd), arsenic (As), manganese (Mn), and chromium (Cr)), and air pollutants (such as volatile organic compounds (VOCs), polyaromatic hydrocarbons (PAHs), particulate matter $(PM)_{10}$ and $PM_{2.5}$, and nitrogen oxides $(NO_x)$). These neurotoxic elements undoubtedly affect neurodevelopment in children. Nonetheless, the results found in this study indicate that further studies are necessary to examine the possible action of Al and other neurotoxicants in ASD. Other than that, the ASD and TD groups lacked comparability for ethnicity, children's gender, and socioeconomic status. This imbalanced distribution between each group of children might cause a biased approximation of the effect of urinary Al on ASD. However, the regression analysis was performed to overcome this issue. Lastly, ASD children recruited in this study were comprised of high-functioning ASD and moderate-functioning ASD but not low-functioning ASD due to the unavailability of this group in Genius Kurnia center, as this institution only recruited children with high-functioning ASD and moderate-functioning ASD for early intervention. If children with low-functioning ASD were included in the study, there would be more valuable findings discovered.

## CONCLUSIONS

Higher urinary Al levels over 1.60 µg/dL were discovered to be a significant risk factor for ASD in preschool children in Kuala Lumpur. It suggests that increased exposure to Al may occur, and that ASD may be associated with the availability and neurotoxic effect of Al in the brain. Future studies are needed to explore the relationships of other clinical samples of Al, such as blood, nails, and hair, with ASD among preschool children in different urban cities in Malaysia.

## ACKNOWLEDGEMENTS

The authors thank the UKM Department of Community Health and the Faculty of Medicine for their assistance in this research. The authors would also like to thank Genius Kurnia, Ministry of Education Malaysia, and Kemas, Ministry of Rural Development Malaysia, for allowing permission to perform this research.

### Funding

This work was supported by the Ministry of Health (MOH) Malaysia Medical Research and Ethics Committee (NMRR-18-3765-45117) and the National University of Malaysia (UKM) Research and Ethics Committee (FF-2018-286). The funders had no role in study design, data collection and analysis, decision to publish, or preparation of the manuscript.

### Grant Disclosures

The following grant information was disclosed by the authors:

Ministry of Health (MOH) Malaysia Medical Research and Ethics Committee: NMRR-18-3765-45117.

National University of Malaysia (UKM) Research and Ethics Committee: FF-2018-286.

## Competing Interests

The authors declare there are no competing interests.

## Author Contributions

- Mohd Shahrol Abd Wahil conceived and designed the experiments, performed the experiments, analyzed the data, prepared figures and/or tables, authored or reviewed drafts of the article, and approved the final draft.
- Mohd Hasni Ja'afar conceived and designed the experiments, performed the experiments, analyzed the data, prepared figures and/or tables, authored or reviewed drafts of the article, and approved the final draft.
- Zaleha Md Isa performed the experiments, analyzed the data, authored or reviewed drafts of the article, and approved the final draft.

## Human Ethics

The following information was supplied relating to ethical approvals (*i.e.*, approving body and any reference numbers):

The Ministry of Health (MOH) Malaysia Medical Research and Ethics Committee (NMRR-18-3765-45117) and the National University of Malaysia (UKM) Research and Ethics Committee (FF-2018-286).

## Field Study Permissions

The following information was supplied relating to field study approvals (*i.e.*, approving body and any reference numbers):

The Ministry of Health (MOH) Malaysia Medical Research and Ethics Committee (NMRR-18-3765-45117) and the National University of Malaysia (UKM) Research and Ethics Committee (FF-2018-286).

## Data Availability

The raw measurements are available in the Supplementary File.

## Supplemental Information

Supplemental information for this article can be found online at http://dx.doi.org/10.7717/peerj.15132#supplemental-information.

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
