# Peer review of "Urinary aluminium and its association with autism spectrum disorder in urban preschool children in Malaysia"

_PeerJ, doi:10.7717/peerj.15132_

## Round 0.1 · original submission · Major Revisions

Thank you for your submission. Based on the reviewers' comments, I suggest major revisions for this manuscript. Please respond to the reviewers' comments.

·

Basic reporting

1. Please follow the reference format suggested by PeerJ. The current reference format is inconsistent as some references contain information like doi link and pdf link (line 338, 389, 412).

2. For Figure 1, do different colors represent the severity of ASD status? What criteria is used to determine the severity of ASD status?

Experimental design

1. Participants for this study were randomly selected from different types of sites (Genius Kurnia center, Tabika Kemas public preschools, and Taska Kemas public nurseries) as case and control groups. Would the author consider any related random effects for sites for modeling?

2. As the author mentioned in the discussion regarding the potential source of aluminum (i.e., water, soil), are there other variables available in the dataset that can be considered in the analysis?

Validity of the findings

1. The author used a nonparametric test (Mann-Whitney U Test) to compare two groups in many cases in this paper. Did the author check the distribution of continuous variables? If so, please mention it in the paper.

2. Any results from the multiple logistic regression that the urinary AI variable is treated as a continuous variable?

3. Multiple variables were adjusted in the logistic regression. Although some variables are statistically significant in the analysis, it is confusing to interpret them as major risk factors for ASD due to the author only considered a limited number of risk factors (i.e., high education level, high-income level).

Additional comments

Echo the question from PeerJ stuff: a better motivation for studying AI and ASD can improve the quality of the paper a lot, and further differentiate from the previous study.

Reviewer 2 ·

Basic reporting

This paper described the association between urinary Aluminum and autism spectrum disorder among 155 preschool children in Malaysia, and argued that Al was a significant risk factor.
This paper was written in clear English, sufficient introduction and clear structure. All tables and figures were labeled.

Minor points:
Line 3: add “in Malaysia” in the title because it only included one country.
Line 207 and 210: repeated same citation in one sentence.
Line 210 and 213: repeated same citation in one sentence.
Line 263-265: add a source for “Many developed …. pollutants”.
Table 1: spell out “RM” at first use.
Figure 1: remove the word “Figures” on top-left.

Experimental design

The research question was meaningful and well-defined.

Major points:
Line 107-110: would be clearer if authors can provide a consort diagram for the number of randomly selected, inclusion/exclusion.

Validity of the findings

Major points:
Line 143-144: Authors mentioned a p-value here, but the authors only provided the p-value for median, and did not provide the p-value for mean income in Table 1.
Line 158 and Figure 1: Mann-Whitney U Test can only test for two groups, how did authors compare three groups by this test in Figure 1 (B)? Please explain.
Line 165: In the dataset authors provided, there were 2 missing values in the categorized Al variable "aluminumcode", which had the original Al values 1.608 and 1.609. Did authors use this "aluminumcode" in the logistic regression analysis (Table 3)? Please explain why they were missing.
Line 168-169: How did authors select or determine these variables as significant risk factors in multiple logistic regression analysis? Please explain the process (from univariate? U test? or manually selected?), or from other literatures. There were other potential risk factors in the dataset, but they were not mentioned in this paper. Please also elaborate on the description of statistical analysis in the method section (for replication).
Line 289-294: Authors showed that smoking was a risk factor for Al here, and I saw there was a smoking variable in the dataset, please explain why smoking (e.g., active/passive) was not considered as a risk factor in the analysis? Again, please elaborate on how you selected or determined variables as significant risk factors in the method section.
Dataset: I am not sure how authors converted the education level from three categories to two categories, the numbers were inconsistent. Please explain.
For example: in “parenedu”, there were 16 “primary education”, 111 “secondary education” and 28 “tertiary education”; but in “pareneduxxx”, there were 57 “secondary education” and 98 “tertiary education”.

Minor points:
Line 172: missed “odds of …” for “2.7 times”
Table 3: Authors used “Race” here, but it was "Ethnicity" in Line 168 and Table 1.

Additional comments

In the method section, authors need to provide more details on the method of statistical analysis. Some outstanding inconsistencies in the dataset need to be explained. The results/tables will need to be reviewed again upon the explanations.

Annotated reviews are not available for download in order to protect the identity of reviewers who chose to remain anonymous.

---

## Round 0.2 · Minor Revisions

Please address the reviewer's comments in addition to mine.

1. In Figure 2, please consider utilizing a scattered box plot to display both the mean and median simultaneously. Alternatively, if you prefer to show the median and mean in separate plots, please include error bars and scattered points on both. To ensure consistency, it is recommended to maintain the same y-axis scale for both figures, as they represent the same data.

2. In Line 131, since it is stated that "The normality of the data was examined graphically using Q-Q plot and histogram," it is expected to see both graphs, but they are not present.

3. In Line 219, with respect to the discourse about establishing a cut-off value for urinary Al level, it should be noted that the proposed cut-off was based on a case study with small sample size, and it has not been validated on an independent dataset. Hence, I do not recommend utilizing this cut-off value for pre-school children residing in the urban city of Kuala Lumpur and list this as a limitation.

4. In Line 287, The case-control study's typical limitations, such as inadequacies in measuring multiple risk factors, sampling bias, and others, are not addressed in the discussion.

·

Basic reporting

My earlier comments and suggestions have been well addressed. No further questions.

Experimental design

My earlier comments and suggestions have been well addressed. No further questions.

Validity of the findings

My earlier comments and suggestions have been well addressed. No further questions.

Reviewer 2 ·

Basic reporting

no comment

Experimental design

no comment

Validity of the findings

no comment

Additional comments

Great improvement! Prior concerns were addressed accordingly.

Minor points:
Line 167-168 and Figure 3: Do the AUC and Figure 3 need to be updated, as the logistics regression model was updated (e.g., adding back 2 missing Al “aluminumcode”)?
Line 308-310: Al may be a risk factor (or association). The results that authors had may not conclude a causal relationship. Should be careful about the wording “be caused by”.

---

## Round 0.3 · accepted · Accept

Thank you so much for addressing all the comments.

Reviewer 2 ·

Basic reporting

No comment.

Experimental design

No comment.

Validity of the findings

No comment.

Additional comments

Prior concerns were addressed accordingly.